# Nudging accurate scientific communication

**Aurélien Allard**, **Christine Clavien***

iEH2-Institute for Ethics History Humanities, University of Geneva, Geneva, Switzerland

* Christine.Clavien@unige.ch

## Abstract

The recent replicability crisis in social and biomedical sciences has highlighted the need for improvement in the honest transmission of scientific content. We present the results of two studies investigating whether nudges and soft social incentives enhance participants' readiness to transmit high-quality scientific news. In two online randomized experiments (Total N = 2425), participants had to imagine that they were science journalists who had to select scientific studies to report in their next article. They had to choose between studies reporting opposite results (for instance, confirming *versus* not confirming the effect of a treatment) and varying in traditional signs of research credibility (large *versus* small sample sizes, randomized *versus* non-randomized designs). In order to steer participants' choices towards or against the trustworthy transmission of science, we used several soft framing nudges and social incentives. Overall, we find that, although participants show a strong preference for studies using high-sample sizes and randomized design, they are biased towards positive results, and express a preference for results in line with previous intuitions (evincing confirmation bias). Our soft framing nudges and social incentives did not help to counteract these biases. On the contrary, the social incentives against honest transmission of scientific content mildly exacerbated the expression of these biases.

## Introduction

Since 2011, scientists in the social and bio-medical sciences have felt a growing unease with the methods used in their disciplines. The so-called replicability crisis has highlighted the limits faced by traditional methods and the traditional publication model. In psychology, only 36% of 97 targeted experiments were successfully replicated [1], compared to a slightly higher replicability rate of 61% in the related field of experimental economics [2]. Beyond social sciences, the recent Replication Project: Cancer Biology found a replicability rate of 46% in preclinical cancer biology research [3].

The transmission of inaccurate information is not limited to the work of scientists, however. Research has shown that newspapers reporting of scientific articles is also biased towards positive results, thus amplifying a preference that is already present at the scientific level [4]. In a broader context, recent years have seen increased attention paid to the phenomenon of fake news and the transmission of unreliable information [5]. Ordinary citizens are important conveyors of scientific information, in their decision to share information with friends, family,

**Data Availability Statement:** All data files are available from the OSF database (https://osf.io/pm2cq/). DOI: 10.17605/OSF.IO/PM2CQ.

**Funding:** Both A.A. and C.C. were supported by the European Union's Horizon 2020 Research and

Innovation Programme [grant no. 824586], https://commission.europa.eu/research-and-innovation_en. The funder had no role in study design, data collection and analysis, decision to publish, or preparation of the manuscript.

**Competing interests:** The authors have declared that no competing interests exist.

and, in the case of social media, even with random strangers. This highlights the need for developing a general framework for predicting the transmission of accurate information, both for researchers and non-researchers. Determining which factors lead to the transmission of accurate information could be central to promote research integrity, both in a teaching context and for the general public.

In this paper, we study several factors influencing non-specialists' treatment of scientific information. We study the influence of two major biases: a positive results bias, and confirmation bias, both of which have been identified as possible causes of the replicability crisis [6, 7]. The bias for positive results has been extensively studied in recent years. Research has shown that journals' editors and peer-reviewers are more likely to recommend the publication of articles reporting positive results [8, 9]. In response, authors adapt to these incentives and tend to file-drawer studies that show negative results. For instance, 65% of experiments with null results from the *Time-sharing Experiments in the Social Sciences* (*TESS*) program were never even written up by their authors, compared to only 4.4% of experiments returning strong evidence in support of the stated hypotheses [10].

Beyond the bias for positive results, we also investigate confirmation bias, or the tendency to look for and transmit information that confirms our own preexisting intuitions [11]. Confirmation bias can entrench mistaken results in the published literature, if failed replications are disregarded by the authors as likely products of errors or unknown confounds [6]. Philosopher Liam Bright even speculated that confirmation bias could lead authors to commit fraud, if they choose to manipulate evidence so that it fits with what researchers consider to be "true" [12].

While it is tempting to see the transmission of information only from the perspective of biased transmission, past research shows that the public can correctly recognize signs of research reliability, such as high sample sizes, the use of randomized control trials, or a high level of prior plausibility [13]. Public understanding of these factors is limited, however; for instance, in a recent poll, only 60% of participants correctly identified the need for a control group to test the effectiveness of a new drug [14].

The first motivation of this paper is diagnostic, as we try to understand positive and negative factors influencing the transmission of (in)accurate information. Our second motivation is practical: we also examine if, with minimal inputs, it is possible to steer people's choices towards a more reliable treatment of information. Our practical goal was to mitigate the influence of the positive results bias and to enhance preferences for more rigorous research. Overall, we considered that an ideal communicator of science would prefer to report information on experiments with a high sample size, with the presence of a control group, and would show no bias towards positive results. To encourage participants towards this ideal, we chose to study the impact of nudges, or minimal interventions that try to influence participants in a desirable direction without changing the incentive structure faced by the participants [15]. We adopt a pragmatic perspective on nudges, as we consider them to be one of the tools that can be used by decision-makers to promote desirable goals, alongside interventions on incentives and strict proscriptions of bad behaviors. Since it may be easier to implement nudges than strict prohibitions or to promote strong changes in the incentives structure, it is important to study the possible benefits of nudges in their own right.

We take our inspiration from two recent nudge interventions that had some success in limiting the transmission of fake news. Pennycook & colleagues [16–18] showed that people can transmit fake news even in a context where they recognize that the news is implausible. However, asking participants beforehand to rate the plausibility of fake news reduces the willingness to transmit them, possibly because it heightens the norm of accuracy in

participants' minds. Similarly, Lisa Fazio showed that asking participants to pause and think about the reliability of new information decreases participants' willingness to share fake news [19]. Both interventions show that it is possible to improve participants' accuracy at transmitting information with subtle reminders, even without the use of intensive interventions.

In study 1, we tested the effect of similar framing procedures. We asked participants to imagine that they were journalists choosing new scientific experiments to report upon. We manipulated the descriptions of the experiments, some being more rigorous than others (large *versus* small number of participants, presence *versus* absence of a control group), and the catchiness of the reported results (null *versus* positive results). We then used two different kinds of soft nudges to steer participants towards the reporting of more rigorous results, and to limit the preferences for positive results.

For our first nudging intervention, we took inspiration from the method of multiple hypotheses promoted by Platt, who hypothesized that questionable research practices stem from an exclusive focus on one specific hypothesis [20]. Platt's hypothesis seems corroborated by research on attentional biases, which highlights the fact that participants tend to neglect possibilities that are not directly presented to their attention (What Nobel laureate Daniel Kahneman summarized as "What you see is all there is", or Wysiati. Kahneman, 2011; Tversky & Koehler, 1994) [21, 22]. By stressing the possibility of obtaining null results, we hoped to make it a live possibility in participants' minds. Our nudging intervention consisted in presenting with equal standing both a positive hypothesis ("Intervention X works") and an associated null hypothesis ("Intervention X does not work"). We call this intervention the *Attention to the Null Hypothesis* nudge. We hoped that this subtle framing of information would positively impact participants' propensity to transmit reliable science.

Our second nudge targeted people's sensitivity to role-attribution, that is, the fact that people tend to adopt rule-based behavior depending on their perception of their own social role. Social theorists have argued that people tend to follow different behavioral scripts depending on how they perceive the social expectations around them [23, 24]. Especially important in this context are social roles, and the different obligations associated with each social function. For instance, a store manager and a customer would respond differently if they observed a theft in the store, and the differences in behavior are rooted in different obligations associated with each person's role. Even 7 years-old children understand that different duties are associated with different social positions [25]. Most importantly for our purpose, social roles are ambiguous: different duties are associated with the same social position. In our case, the social duties of journalists depend on the types of social relationships in which their employment is embedded. As purveyors of information to the public, they have a duty to be accurate and to avoid any kind of deception. As employees in capitalist firms, they have a duty to be interesting so as to stimulate sales and profit. In our second nudging intervention, we manipulated the salience of different social roles associated with journalism, by either emphasizing the importance of transmitting accurate results, or of writing interesting salable papers. We call this intervention the *Social Role Nudge*. We hoped that by stressing the importance of reliable information, participants would be more likely to report reliable scientific results.

In Study 2, we study whether it is possible to steer people towards reporting more rigorous research by mixing social influence with traditional economic incentives. We asked participants to imagine that they were pressured either by a colleague or a boss towards reporting salesworthy (vs high-quality) research. This combined manipulating social expectations (as in Study 1) with the added possibility of economic sanctions if the participant did not conform to the social pressure.

## Study 1

### Methods

We received written ethical approval from the University of Geneva's Committee for Ethical Research (CUREG -2020-12-20). All participants provided informed consent by ticking a box before answering our questions. By design, we had no access to sensitive or personal data (health status, name, e-mail, etc.) about participants. Both studies 1 and 2 were pre-registered. The pre-registrations can be found at https://osf.io/d4fet/ (Study 1) and https://osf.io/q2ck6/ (Study 2). Throughout the manuscript, we report all manipulations and probes that were conducted. Full materials, data, and R code used to analyze the data can be found at https://osf.io/pm2cq/.

**Participants.** We recruited 1122 American residents on Amazon Mechanical Turk [26] in January 2021. We only recruited participants with over 95% HIT accuracy, indicating that participants have successfully completed at least 95% of their tasks on Amazon Mechanical Turk. Participants were paid $0.80 for a 4 minutes task. We expected to reject a large number of participants after the application of quality checks, and aimed to have a final sample size of around 800 participants.

The sample size choice was led by considerations of statistical power, tempered by resource constraints. We tried to estimate two main effect sizes of interest: the impact of study features (such as the use of a control group, or the existence of positive results) and the interaction between study features and nudges. We performed simulations in R to estimate our statistical power (See the *R* code on the associated OSF page). We simulated our experimental design, with "participants" randomly assigned to each condition present in our study. We varied two kinds of effects: a main effect of study feature (*Positive results*, *RCT*, and *Sample size*), and an interaction effect between study features and experimental condition. For simplicity, the dependent variable (preference for the first experiment seen by the participant) was modeled as a normal random variable, with a residual standard deviation of 1 and a mean depending on the impact of study features and experimental effects. We found that a sample size of 800 was sufficient for obtaining 80% statistical power to detect even a small effect of study features, corresponding to a Cohen's d of 0.2. In the case of the *Attention to the Null hypothesis* intervention, where we had only two experimental conditions, 800 participants also gave us 80% power to detect moderate effect size interactions (corresponding to an increase of 0.4 in the standardized impact of study features). However, in the case of the *Social Role* nudge, where we chose to select four experimental conditions, the same sample size only gave us 51% power to detect moderate effect size interactions, with adequate (80%) power only to detect large effect sizes corresponding to a standardized effect size of 0.56. Our budget constraints prevented us from increasing the sample size, but we still considered it important to estimate the effect size of our *Social Role* nudges. Due to the possibility of false negatives, we consequently remain cautious in our interpretation of our results.

**Materials.** We asked participants to imagine that they were journalists writing an article about scientific research [inspired by 27]. In the preparation of their next article, participants had to read about two scientific experiments and had to indicate how they would incorporate them in their article. Participants were randomly assigned to reading stories about one out of five kinds of scientific studies: studies could test the impact of 1) a new medical drug, 2) a new psychological therapy, 3) having a growth mindset, 4) a Critical Thinking training, and 5) microfinance on poverty.

The two experiments varied on three main factors: the use of a control group (*No control group vs Randomized control trial*), the sample size (*Low vs High*), and the type of results (*Positive vs Null results*). Variation in each of the three factors was orthogonal; the first experiment

had a 50% chance to be the low sample size experiment, for instance, but this had no impact on whether it would also be the *Randomized Control Trial* (Henceforth: *RCT*), or the *Positive result* experiment.

We also randomly varied the university of the researchers (*Yale vs Princeton*) to make the vignettes look more plausible by creating some differences between the vignettes, but this was not a variable of interest.

For instance, in the new medical drug condition, the vignettes read as follows (most important randomized elements in bold):

*Researchers from Princeton university have recruited 200 participants [**High sample size**] from a local hospital because they had hypertension. All participants were given the new drug [**No RCT**]. After 5 days, the situation of 70% of participants had improved, since their blood pressure decreased; the situation of 20% of participants stayed the same, and the blood pressure of 10% of participants increased. The researchers concluded that the drug was working [**Positive result**].*

*Researchers from Yale university have recruited 40 participants [**Low sample size**] from a local clinic because they had hypertension. Half of the participants were given the new drug and half a placebo [**RCT**]. After 5 days, around 50% of the participants had seen their condition improve in both the placebo group and the treatment group, since their blood pressure decreased. The researchers concluded that the drug was not working [**Null result**].*

In this example, each experiment possessed different attributes of rigor: the first experiment had a high sample size, but did not have a control group, while the second experiment had a low sample size, but included a control group. In this case, we make no *a priori* prediction about which experiment should be preferred by participants. 50% of participants saw a similar vignette with a conflict between different kinds of rigor. However, since *Sample size* was manipulated independently of *RCT*, 50% of participants saw a vignette where the same experiment included both a high sample size and the presence of a control group. In this case, participants should prefer to report on the experiment with both a high sample size and a control group.

Here is an example where participants had to choose between an experiment with both signs of rigor present and an experiment with a very low level of rigor (in the *Critical Thinking Training* vignette):

*Experiment A: Researchers from Yale University have recruited 40 undergraduate students [**Low sample size**] and submitted them to a Critical Thinking program. They measured how many fake news participants shared on Twitter. They found that students shared less fake news after the intervention compared to before the intervention [**No RCT**]. The researchers concluded that the intervention was working [**Positive result**].*

*Experiment B: Researchers from Princeton University have recruited 200 undergraduate students [**High sample size**] in a Critical Thinking program. Half of them participated in a fake news training program, and half of them were kept as a control group and received no training [**RCT**]. They measured how many fake news participants shared on Twitter. They found that participants were equally likely to share fake news in both the control group and the training group. The researchers concluded that the intervention was not working [**Null result**].*

In this case, the positive result was found in the experiment with the lowest level of rigor. However, since the positive result was manipulated independently of *RCT* and *Sample size*,

participants were equally likely to find the positive result in the experiment with a high level of rigor. Other examples of vignettes can be found in S1 Appendix in S1 File.

After reading the vignette, participants had to indicate how much weight they would put on each of the experiments in their own newspaper article. This was the main dependent variable of our experiment.

Participants' choice read as follows:

*I would only report the results of experiment A.*

*I would report on both experiments, but I would put more emphasis on experiment A.*

*I would report on both experiments, and would put equal emphasis on both.*

*I would report on both experiments, but I would put more emphasis on experiment B.*

*I would only report the results of experiment B.*

We then recoded these answers to constitute a *Preference for the first experiment* variable. We constituted a numerical variable ranging from 1 to 5, with 1 corresponding to "I would only report the results of experiment B" and 5 corresponding to "I would only report the results of experiment A".

On the following page, we asked participants to justify their choice with an open-ended answer. We used their answer to exclude inattentive participants (see below).

Before participants read the experiments and made their choice, however, we randomly assigned them to different experimental conditions. In a 2*4 factorial design, we randomly manipulated the kind of hypothesis participants reported upon (*Positive hypothesis only vs competing hypotheses*) and the mission assigned to participants (*Reporting rigorous vs interesting results vs improving the world vs control*).

*First intervention*: *Attention to the null hypothesis nudge*. Before reading the specific scientific studies, we assigned participants to read different descriptions of the hypotheses that scientists were testing. In one experimental condition (*Positive hypothesis only*), participants had to report their initial intuition concerning the hypothesis that they were tasked to assess (e.g., they had to indicate how plausible it was that a new drug was effective in treating some illness). In the other experimental condition (*Competing hypotheses*), we asked participants to report on the plausibility of two competing hypotheses: the hypothesis that the intervention had a positive impact, and the hypothesis that the intervention had no impact.

For instance, in the case of the medical drug, the description of the hypotheses was as follows:

*You have chosen to cover a new drug, Xoliphenon, that has been invented to improve treatment of hypertension.*

*[Positive] You are writing this article to report on the following scientific hypothesis: Xoliphenon is effective at reducing blood pressure.*

*[Competing] You are conducting this research to see how research stands between two opposing scientific hypotheses: A) Xoliphenon is effective at reducing blood pressure, B) Xoliphenon has no impact on blood pressure.*

Before reading the experiments, participants then had to give their initial intuition concerning the plausibility of the positive hypothesis (*Positive hypothesis only* condition) or of both hypotheses (*Competing hypotheses* condition). For instance, in the *Competing hypotheses*

condition, the probe read as follows: "What is your intuition regarding these hypotheses?", and participants had the choice between the following five different options:

- Hypothesis A is definitely true.

- Hypothesis A is probably true.

- Both hypotheses are equally likely to be true.

- Hypothesis B is probably true.

- Hypothesis B is definitely true.

In the *Positive hypothesis only*, these options ranged from "This hypothesis is definitely true" to "This hypothesis is definitely false" (see full text in S1 Appendix in S1 File).

*Second intervention*: *Social role nudge*. Before reading the two experiments, we tried to nudge participants towards different social roles as journalists. The participants could read:

- *While reading these studies, please keep in mind that your goal as a journalist is...*

- *to have a positive impact on the world.*

- *to publish the most interesting article.*

- *to give an account of research that is as accurate as possible.*

We set no specific goal to participants in the control condition.

We expected participants in the *Most interesting* condition to have higher preferences for positive results compared to participants in the control condition, and participants in the *Accurate* condition to be more influenced by signs of research quality, and less influenced by the existence of positive results, compared to the control condition. We had no specific intuition regarding the *Positive impact* condition, and implemented it for exploratory purposes.

*Other probes*. We also collected the following personality and cognitive variables for exploratory purposes: *Faith in intuition*, *Need for evidence*, and three items on science understanding. *Faith in intuition* was measured with items like "I trust my initial feelings about the facts", with agreement on a labeled 1 to 5 scale going from "Disagree strongly" to "Agree strongly". Both the *Faith in intuition* scale and the *Need for evidence* scale were shortened versions of the scales used in Garrett and Weeks (2017) [28]. The science understanding items were newly developed for this study and included items intended to measure the understanding of experimental methods, such as "To measure the impact of an intervention, it is essential to compare two groups: one with, and one without, the intervention". Full probes for all exploratory scales can be found in S1 Appendix in S1 File. We also asked participants to report their age, gender, and education level. *LimeSurvey* also collected the participants IP address, which we later used to exclude participants with dubious IP addresses (see below). Since the IP addresses could be used to identify participants, we later deleted this variable from our records.

## Statistical analysis

We used linear regressions to predict participants' preference for the first experiment they saw. While our dependent variable is strictly speaking an ordinal variable, we used linear regression for simplicity, in conformity with past experiments on a similar topic [16, 27]. We also feel partially justified in this choice by the fact that recent research has shown a roughly linear impact of psychological ordinal-variable scales on real-life behaviors [29].

All analyses were performed with the help of the R software version 4.3.0, Rstudio, and the following packages: tidyverse, papaja, and gtsummary [30–34].

**Table 1. Demographic information for Study 1.**

| Characteristic | N = 736 |
|---|---|
| **Gender** | |
| Female | 40% |
| Male | 60% |
| Other | 0.4% |
| **Age** | 39 (12) |
| **Education** | |
| No higher education | 0.3% |
| High school degree | 25% |
| Undergraduate degree | 60% |
| Master Degree, PhD Degree, or Professional degree (M.D.,…) | 15% |

## Results

**Participants exclusion.** In light of recent concerns with Mturk data quality [35], we applied three different exclusion criteria. First, we excluded participants who did not give any coherent justification. This concerns almost exclusively participants who did not write any sentence, and a subset of participants who provided nonsensical text, including obviously copy-pasted citations (e.g. "Elements of Bader's theory of atoms in molecules are combined with density-functional theory to provide an electron-preceding perspective on the deformation of materials"). This exclusion was based solely on the justification the participants offered and was blind to the other answers provided by participants. Second, we excluded participants whose IP address indicated that they were probably not based in the United States [36]. Third, we excluded participants whose Mahalanobis distance on the personality items was higher than the 95% percentile of a chi-squared distribution with 7 degrees of freedom, 7 being the number of items in our personality scales [37]. While outliers exclusion methods traditionally exclude participants higher than the 99.9% percentile, we felt that excluding more participants was needed to prevent the inclusion of inattentive participants. All three exclusion methods were pre-registered. Following the application of our three exclusion methods, we were left with 736 participants, out of the 1122 initial participants. We provide the full demographic statistics for our final sample in Table 1.

**First model: Predicting experiment choice based on study features.** In our first model, we estimate the preference for the first experiment seen by the participants, depending on whether the first experiment uses a control group, has a high sample size, and reports positive results. As seen in Table 2, in conformity with our predictions and the results of previous studies, all three predictors are significant and show strong effect sizes. Participants display a preference for experiments using randomization ($b = 0.61$, $p < .001$), for higher sample sizes ($b = 0.35$, $p < .001$), and for studies reporting positive results ($b = 0.41$, $p < .001$).

**Table 2. Predicting preference for first experiment based on methodological features and presence of positive results.**

| Predictor | $b$ | 95% CI | $t(732)$ | $p$ |
|---|---|---|---|---|
| Intercept | 2.32 | [2.19, 2.45] | 34.44 | $< .001$ |
| RCT | 0.61 | [0.48, 0.74] | 9.03 | $< .001$ |
| High Sample Size | 0.35 | [0.22, 0.48] | 5.18 | $< .001$ |
| Positive Results | 0.41 | [0.28, 0.54] | 6.07 | $< .001$ |

**Second model: Predicting experiment choice based on methodological features, presence of positive results, and nudges.** In our second model, we keep the same three predictors as in our first model, but include interactions with our two kinds of nudges, the *Attention to the Null Hypothesis* nudge and the *Social Role* nudge. We predicted that the *Attention to the Null Hypothesis* nudge would increase the preference for RCT and high sample size, and would decrease the preference for positive results. In statistical terms, this would correspond to a positive interaction between *Positive Hypothesis Only* and *Positive Results*, to a negative interaction between *Positive Hypothesis Only* and *RCT*, and to a negative interaction between *Positive Hypothesis Only* and *High sample size*. We similarly predicted that the *Social Role*: *Accuracy* nudge would increase the preference for *RCT* and *High Sample Size*, and decrease the preference for *Positive Results*. On the other hand, we predicted that the *Social Role*: *Interest* nudge would decrease the preference for *RCT* and *High Sample Size*, and would increase the preference for *Positive Results*. As seen in Table 3, and contrary to our predictions, none of the interactions are significant (all p > .08). While the confidence intervals include upper bounds of estimates that could be interpreted as important effects, results are inconsistent, with some point estimates going in the predicted direction, and some point estimates going in the direction opposite to our predictions. For instance, setting the goal as being interesting is (nonsignificantly) associated with a greater preference for positive results, which could be read as a weak confirmation of our hypothesis. However, setting the goal to being accurate is also (nonsignificantly) associated with a greater preference for positive results, which is utterly incompatible with our hypotheses. In the latter case, the lower end of the confidence interval is -0.11

**Table 3. Predicting preference for first experiment based on methodological features (sample size, randomization), positive results, and framing nudges (presenting competing hypotheses on equal footing & attributing different social roles).**

| Predictor | b | 95% CI | t(716) | p |
|---|---|---|---|---|
| Intercept | 2.18 | [1.86, 2.49] | 13.67 | < .001 |
| Positive Results | 0.21 | [-0.11, 0.53] | 1.28 | .199 |
| Positive Hypothesis Only | 0.19 | [-0.08, 0.46] | 1.41 | .159 |
| Social Role: Impact | 0.12 | [-0.27, 0.52] | 0.60 | .550 |
| Social Role: Interest | 0.10 | [-0.29, 0.49] | 0.48 | .630 |
| Social Role: Accuracy | -0.05 | [-0.43, 0.33] | -0.25 | .801 |
| RCT | 0.72 | [0.40, 1.04] | 4.45 | < .001 |
| High Sample Size | 0.44 | [0.13, 0.76] | 2.74 | .006 |
| *Positive Hypothesis Only* | | | | |
| Positive Hypothesis Only × Positive Results | 0.03 | [-0.24, 0.30] | 0.24 | .812 |
| Positive Hypothesis Only × RCT | -0.03 | [-0.30, 0.24] | -0.19 | .847 |
| Positive Hypothesis Only × High Sample Size | -0.17 | [-0.44, 0.10] | -1.22 | .224 |
| *Social Role*: *Accuracy* | | | | |
| Social Role: Accuracy × Positive Results | 0.27 | [-0.11, 0.65] | 1.39 | .164 |
| Social Role: Accuracy × RCT | -0.02 | [-0.40, 0.36] | -0.09 | .925 |
| Social Role: Accuracy × High Sample Size | 0.01 | [-0.37, 0.39] | 0.06 | .955 |
| *Social Role*: *Interest* | | | | |
| Social Role: Interest × Positive Results | 0.29 | [-0.10, 0.67] | 1.47 | .142 |
| Social Role: Interest × RCT | -0.34 | [-0.72, 0.05] | -1.73 | .084 |
| Social Role: Interest × High Sample Size | 0.06 | [-0.33, 0.44] | 0.30 | .765 |
| *Social Role*: *Impact* | | | | |
| Social Role: Impact × Positive Results | 0.09 | [-0.30, 0.48] | 0.45 | .654 |
| Social Role: Impact × RCT | -0.01 | [-0.41, 0.38] | -0.06 | .953 |
| Social Role: Impact × High Sample Size | -0.05 | [-0.45, 0.35] | -0.26 | .797 |

(in raw effect sizes, on a 1 to 5 scale), thus indicating that the nudge could not have any strong impact on accurate information transmission.

To further assess the robustness of this null result, we also conducted additional sensitivity analyses. We estimated the probability of failing to obtain a single positive result if all our nudges had a small positive effect of 0.2 standardized mean difference in the predicted direction. If this were the case, then we would expect to find at least one positive result 89% of the time (see the R code at the associated OSF page). We can thus rule out the existence of a small consistent effect. Overall, these results suggest that our instructions did not strongly affect the already existing preferences for high sample sizes, randomization, and positive results.

**Exploratory analyses: Studying the impact of confirmation bias.** To study the possible impact of confirmation bias, we assess the impact of believing in the truth of an hypothesis on the preference for positive results supporting this hypothesis. We re-coded agreement with the positive hypothesis in a -2 to 2 scale, -2 corresponding to finding the positive hypothesis almost certainly false, 2 corresponding to finding it almost certainly true, and 0 corresponding to finding it equally likely to be false or true. As seen in Table 4, preference for positive results was general, even among participants who judged the hypothesis to be equally likely to be false or true (as seen with the coefficient for Positive Results, $b = 0.21$, $p = .054$). Moreover, the preference for positive results increased among participants who believed the hypothesis to be true, as seen in the significant interaction between positive results and pre-existing belief in the truth of the hypothesis ($b = 0.25$, $p = .010$), indicating the effect of confirmation bias. Both results are suggestive, but the p-values are borderline non-significant in both cases (i.e., close to or above the 0.05 threshold). We therefore replicate these results in Study 2.

In a further exploratory model, we examined whether confirmation bias was reinforced by having a strong faith in intuition (as opposed to basing one's beliefs on evidence). After including the *Faith in intuition* scale in our model, the interaction between belief in the hypothesis, positive results, and *Faith in intuition* was non-significant; the effect was, however, in the predicted direction ($b = 0.12$, $p = 0.31$).

**Non-preregistered robustness check: Excluding the *New medical drug* vignette.** During the peer-review process, one reviewer noticed that one of our vignettes contained a typo. 50% of participants in the *New medical drug* vignette saw the following sentence: "around 50% of the participants had seen their condition improve in both the control group and the placebo group, since their blood pressure decreased. The researchers concluded that the drug was not working". In the first sentence, "control group" should have been "treatment group". We feel that participants would have correctly interpreted this reference to the "control" group as a typo, and that they would correctly have concluded that there was no difference between the treatment group and the placebo group. We thus believe that the main results reported here are not affected. However, we have run all our analyses in both Study 1 and Study 2 after excluding the *New medical drug* condition, and we report the full analysis in S1 Appendix in S1 File. None of the main results are changed by the exclusion of the *New medical drug* condition.

**Table 4. Predicting preference for first experiment based on methodological features, positive results, and agreement with the positive hypothesis.**

| Predictor | b | 95% CI | t(685) | p |
|---|---|---|---|---|
| Intercept | 2.41 | [2.23, 2.59] | 26.44 | < .001 |
| Positive Results | 0.21 | [0.00, 0.42] | 1.93 | .054 |
| Belief in the hypothesis | -0.07 | [-0.22, 0.07] | -1.04 | .300 |
| RCT | 0.58 | [0.44, 0.71] | 8.32 | < .001 |
| High Sample Size | 0.33 | [0.20, 0.47] | 4.78 | < .001 |
| Positive Results × Belief in the hypothesis | 0.25 | [0.06, 0.44] | 2.59 | .010 |

## Discussion

In Study 1, we found that participants showed a preference for reporting practices associated with epistemic credibility, such as high sample sizes and the use of control groups. However, we confirmed that people are vulnerable to confirmation bias and to positive results bias in the reporting of scientific experiments: our participants preferred experiments showing positive results over experiments reporting null results, and preferred experiments confirming their pre-existing beliefs. Moreover, our two nudges failed to influence their preferences. We found this failure of the *Social Role* manipulation to be especially surprising since our manipulation was quite explicit. This result led us to design Study 2, in which we test whether stronger interventions, including social pressure and classical economic incentives, could lead participants to shift towards reporting more accurate results.

## Study 2

Since the soft framing nudges used in study 1 did not significantly impact participants' choices, we decided to test the effect of less subtle incentives. Study 1 showed that social norms *per se* might not have a major effect on the transmission of reliable information. However, we could expect an increased effect when social norms are combined with economic incentives (e.g. fear of being fired if someone is not conforming to the social culture).

In study 2 we test whether the social incentive of peer-culture and top-down pressure has an effect on the honest transmission of scientific information. Since such forms of social incentives can be positive or negative for trustworthy information transmission, we test the case of both a pro-science work culture and pro-business work culture. More precisely, we asked participants to imagine that they had only recently started their job as journalists, and that they were pressured towards promoting either accurate scientific research or salesworthy research, by either a knowledgeable colleague or their boss. Since participants could presumably understand the risks of being fired in case of failing to adapt to the work culture, our manipulation moved beyond the realm of pure (incentive-less) nudges to a domain where participants could reasonably understand the direction of their economic interests, and could thus transmit reliable information out of self-interest.

The second goal of Study 2 was to replicate the effects we found in Study 1. We kept the same features in describing each experiment (presence *vs* absence of a control group, differences in sample size, presence *vs* absence of a positive result) to see if we could replicate the preference for more rigorous experiments and for positive results. We also kept the *Attention to the Null Hypothesis* manipulation, in order to see if we could replicate the null result found in Study 1. However, we dropped the *Social Role* manipulation, as we found it to be too close to the *Social Pressure* manipulation.

### Methods

**Participants.** We recruited 1303 UK participants on *Prolific Academic* [38] in May 2021. Participants were paid £0.80 for their participation. We used the same exclusion methods as in Study 1. This led to the exclusion 143 participants, resulting in a final sample size of 1160 participants.

**Materials.** We used the same materials as in Study 1, varying only the experimental conditions. We kept the *Attention to the Null Hypothesis* nudge from Study 1; that is, participants were randomly attributed to either a *Positive hypothesis only* or a *Competing hypotheses* condition. However, we did not use the *Social role* nudge from Study 1, and replaced it with a *Social pressure* intervention. For the *Social pressure* manipulation, participants were randomly attributed to five different experimental conditions: a control condition, and four experimental

conditions, where we orthogonally varied two factors: whether participants were pressured towards promoting accurate journalism or salesworthy journalism (*direction of pressure*), and whether they were pressured by their boss or by a knowledgeable colleague (*origin of pressure*).

The *colleague* conditions read as follows:

*Please imagine that you are a journalist, who recently started working for an online magazine. During your first day at your job, you are mentored by a successful journalist, who has been working here for 10 years. He gives you the following advice: 'Here, we are trying to boost sales. My advice would be to select stories that are most likely to captivate the readers.' ['Here, we are trying to promote high-quality journalism. My advice would be to select stories that are most likely to be accurate.']*

The *boss* conditions read as follows:

*Please imagine that you are a journalist, who recently started working for an online magazine. During your first day at your job, your boss made it clear that you had to promote the information most likely to boost sales [highest quality information]. He told you to promote the most captivating stories [most accurate stories].*

As specified in our preregistration, our main variable of interest was the direction of the pressure (*accuracy* vs *saleworthiness*), and we manipulated the source of the pressure for exploratory purposes.

## Results

The full demographic information for participants in Study 2 is reported in Table 5.

**First model: Predicting experiment choice based on study features.** In our first model, we estimate the preference for the first experiment seen by the participants, depending on whether the first experiment uses a control group, whether it has a high sample size, and based on whether it reports positive results. Replicating the results of our first experiment, all three predictors are significant and show quite strong effect sizes (Table 6). Participants show a preference for higher sample sizes ($b = 0.36$, 95% CI [0.26, 0.45], $t(1156) = 7.27$, $p < .001$), for experiments using randomization ($b = 0.26$, 95% CI [0.17, 0.36], $t(1156) = 5.40$, $p < .001$), and for studies reporting positive results ($b = 0.49$, 95% CI [0.39, 0.58], $t(1156) = 9.98$, $p < .001$).

**Second model: Predicting experiment choice based on experimental features and nudges.** In our second model, we keep the same three predictors as in our first model, but

**Table 5. Demographic information for Study 2.**

| Characteristic | N = 1,185 |
|---|---|
| **Gender** | |
| Female | 65% |
| Male | 35% |
| Other | 0.7% |
| **Age** | 38 (13) |
| **Education** | |
| No higher education | 3.6% |
| High school degree | 34% |
| Undergraduate degree | 45% |
| Master Degree, PhD Degree, or Professional degree (M.D.,...) | 18% |

**Table 6. Predicting preference for first experiment based on methodological features and positive results.**

| Predictor | b | 95% CI | t(1156) | p |
|---|---|---|---|---|
| Intercept | 2.47 | [2.37, 2.57] | 49.59 | < .001 |
| RCT | 0.26 | [0.17, 0.36] | 5.40 | < .001 |
| High Sample Size | 0.36 | [0.26, 0.45] | 7.27 | < .001 |
| Positive Results | 0.49 | [0.39, 0.58] | 9.98 | < .001 |

include interactions with our two interventions. The results of our *Attention to the Null Hypothesis* nudge are slightly more complicated than the results found in Study 1, since we do obtain one significant result (Table 7). Framing the focal hypothesis solely in terms of the positive hypothesis ("Intervention X has a positive impact") was associated with a greater preference for positive results ($b = 0.20$, 95% CI [0.01, 0.39], $t(1144) = 2.04$, $p = .041$). However, framing the hypothesis testing solely in terms of the positive hypothesis did not significantly decrease the preference for high sample sizes and RCT (all p > .12; the effects were in the predicted direction, however). Given these mixed results, the relatively high p-value (close to the 0.05 threshold), and the multiplicity of tests, caution is warranted. While more research is needed on this topic, given the null results found in Study 1, we expect any possible effect to be small in any case.

To test the impact of our *Social pressure* intervention, to increase statistical power (as specified in our pre-registration), we merged the impact of the *Boss* and *Peer* conditions, to obtain three different conditions: A *control* condition, *Pressure towards quality*, and *Pressure towards Salesworthiness*. Pressure towards quality did not significantly increase the preference for RCT, high sample sizes, or null results (all p > .25; all results in the predicted direction, however. See Table 7 and Fig 1). Pressure towards salesworthiness did not significantly decrease

**Table 7. Predicting preference for first experiment based on methodological features, positive results, and study interventions (*Attention to the Null Hypothesis nudge & social pressure*).**

| Predictor | b | 95% CI | t(1144) | p |
|---|---|---|---|---|
| Intercept | 2.61 | [2.42, 2.80] | 27.24 | < .001 |
| Positive Results | 0.26 | [0.07, 0.45] | 2.67 | .008 |
| Positive Hypothesis Only | 0.07 | [-0.12, 0.26] | 0.70 | .482 |
| Pressure towards Quality | -0.13 | [-0.37, 0.11] | -1.06 | .287 |
| Pressure towards Sales | -0.40 | [-0.64, -0.17] | -3.34 | .001 |
| RCT | 0.30 | [0.11, 0.49] | 3.11 | .002 |
| High Sample Size | 0.31 | [0.12, 0.50] | 3.23 | .001 |
| *Positive Hypothesis Only* | | | | |
| Positive Hypothesis Only × Positive Results | 0.20 | [0.01, 0.39] | 2.04 | .041 |
| Positive Hypothesis Only × RCT | -0.15 | [-0.34, 0.04] | -1.55 | .122 |
| Positive Hypothesis Only × High Sample Size | -0.12 | [-0.31, 0.07] | -1.26 | .209 |
| *Pressure towards Quality* | | | | |
| Pressure towards Quality × Positive Results | -0.09 | [-0.33, 0.14] | -0.78 | .434 |
| Pressure towards Quality × RCT | 0.06 | [-0.17, 0.30] | 0.53 | .598 |
| Pressure towards Quality × High Sample Size | 0.14 | [-0.10, 0.37] | 1.14 | .257 |
| *Pressure towards Sales* | | | | |
| Pressure towards Sales × Positive Results | 0.49 | [0.26, 0.72] | 4.14 | < .001 |
| Pressure towards Sales × RCT | 0.08 | [-0.15, 0.31] | 0.69 | .493 |
| Pressure towards Sales × High Sample Size | 0.19 | [-0.04, 0.42] | 1.64 | .101 |

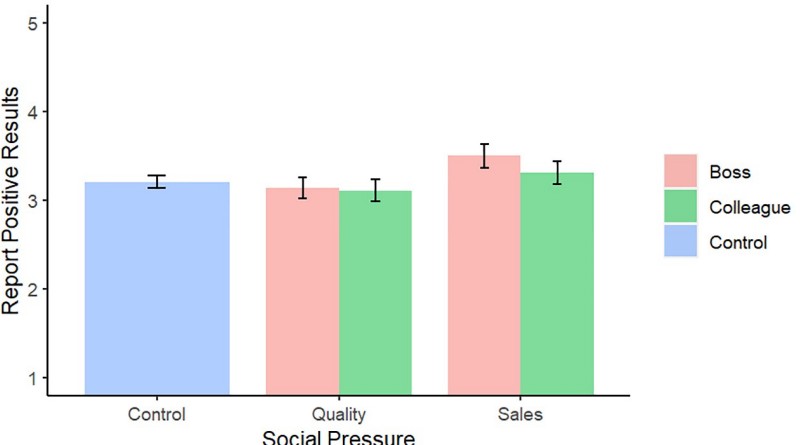

**Fig 1. Preference for positive results depending on social pressure (Study 2).** The y axis represents participant's preference for reporting on the experiment showing a positive result (with 1 indicating that they would not mention this study at all, and 5 that they would only mention this study).

the preference for RCT or high sample sizes (all p > .1, and in the opposite of the predicted direction). However, pressure towards salesworthiness did significantly increase the preference for positive results ($b = 0.49$, 95% CI [0.26, 0.72], $t(1144) = 4.14$, $p < .001$). Most importantly, this result is robust to correction for multiple hypotheses: our p-value adjusted with the Bonferroni correction is 0.0006 if one takes into account all statistical tests performed in the regression and thus remains highly significant.

**The impact of confirmation bias.** To study the impact of confirmation bias, we used the same linear model as in Study 1 (with agreement with the positive hypothesis coded from -2 to 2, 2 indicating the highest degree of belief in the truth of the hypothesis). We fully replicate the results of Study 1: preference for positive results was present, even among participants who judged the hypothesis to be equally likely to be false or true (See Table 8. Statistics for *Positive Results*: $b = 0.37$, 95% CI [0.25, 0.50], $t(1100) = 5.98$, $p < .001$). Also as found in Study 1, preference for positive results was increased among participants who believed the hypothesis to be true, as seen in the significant interaction between positive results and pre-existing belief in the truth of the hypothesis ($b = 0.29$, 95% CI [0.15, 0.43], $t(1100) = 4.13$, $p < .001$).

We also performed the same exploratory model as in Study 1 to further examine whether confirmation bias was reinforced by having stronger *Faith in intuition*. We obtained similar results as in Study 1: the interaction between belief in the hypothesis, positive results, and *Faith in intuition* was non-significant ($b = 0.12$, p = .25), but the effect was in the predicted direction.

**Table 8. Predicting preference for first experiment based on methodological features, positive results, agreement with the hypothesis.**

| Predictor | b | 95% CI | t | df | p |
|---|---|---|---|---|---|
| Intercept | 2.54 | [2.43, 2.66] | 44.26 | 1100 | < .001 |
| Positive Results | 0.37 | [0.25, 0.50] | 5.98 | 1100 | < .001 |
| Belief in the hypothesis | -0.13 | [-0.22, -0.04] | -2.70 | 1100 | .007 |
| RCT | 0.25 | [0.15, 0.35] | 5.03 | 1100 | < .001 |
| High Sample Size | 0.34 | [0.24, 0.43] | 6.84 | 1100 | < .001 |
| Positive Results × Belief in the hypothesis | 0.29 | [0.15, 0.43] | 4.13 | 1100 | < .001 |

**Table 9. Predicting preference for first experiment based on methodological features, positive results, and type of pressure.**

| Predictor | b | 95% CI | t(751) | p |
|---|---|---|---|---|
| Intercept | 2.63 | [2.37, 2.89] | 19.97 | < .001 |
| Positive Results | 0.25 | [-0.01, 0.51] | 1.90 | .057 |
| Pressure towards Sales | -0.43 | [-0.80, -0.06] | -2.30 | .021 |
| Pressure Type: Boss | -0.24 | [-0.60, 0.12] | -1.30 | .193 |
| RCT | 0.19 | [-0.07, 0.44] | 1.42 | .155 |
| High Sample Size | 0.25 | [-0.01, 0.51] | 1.91 | .057 |
| Positive Results × Pressure towards Sales | 0.42 | [0.06, 0.78] | 2.28 | .023 |
| Positive Results × Pressure Type: Boss | 0.00 | [-0.35, 0.35] | 0.01 | .994 |
| Pressure towards Sales × Pressure Type: Boss | 0.31 | [-0.19, 0.82] | 1.22 | .225 |
| Pressure towards Sales × RCT | 0.14 | [-0.22, 0.49] | 0.75 | .452 |
| Pressure towards Sales × RCT | 0.20 | [-0.15, 0.56] | 1.14 | .255 |
| Pressure towards Sales × High Sample Size | 0.35 | [-0.01, 0.71] | 1.93 | .054 |
| Pressure towards Sales × High Sample Size | 0.27 | [-0.08, 0.62] | 1.50 | .135 |
| Positive Results × Pressure towards Sales × Pressure Type: Boss | 0.34 | [-0.16, 0.84] | 1.34 | .181 |
| Pressure towards Sales × Pressure Type: Boss × RCT | -0.25 | [-0.75, 0.24] | -1.01 | .314 |
| Pressure towards Sales × Pressure Type: Boss × High Sample Size | -0.57 | [-1.06, -0.07] | -2.25 | .025 |

**Exploratory model: Exploring the different kinds of pressure.** In a further exploratory model, we perform a linear regression to test whether our *Social pressure* intervention has a stronger impact when the pressure comes from the boss rather than from the colleague. To test this effect, we drop the "Control" condition to analyze the interactions between the direction of the pressure (towards sales *vs* quality), the source of the pressure (boss *vs* colleague), and the features of the experiment (RCT, Sample size, and Result type). Since pressure from the boss could be seen as the strongest kind of pressure, it could be expected that having pressure from one's boss would lead to stronger effects than peer-pressure. Our results do not support this expectation, however; only one of the effects is significant, and in the opposite of the predicted direction, this result being a likely false-positive (pressure from the boss towards sales leading to a higher preference for high sample sizes, p < .025. See Table 9). Regarding the preference for positive results in the case of pressure towards sales (Which was the only significant effect in the previous analyses), the effect is in the predicted direction (pressure from the boss leading to a stronger preference for positive results), but is not significant (p = .18).

## Discussion

Replicating the results of Study 1, participants showed a strong preference for randomized experiments, high sample sizes, and positive results regardless of the experimental condition. Also fully replicating the results of Study 1, we found that participants show a preference for experimental results confirming their prior beliefs, thus evincing confirmation bias. Partially replicating the results of Study 1, we did not find a strong impact of our *Attention to the Null Hypothesis* nudge on participants reporting behavior. However, we found evidence suggesting that putting null and positive hypotheses on an equal footing might lead to a decrease in preference for positive results. These results are, however, tentative.

In a new experimental intervention, we asked participants to imagine that they faced social pressure and social incentives towards either accurate or salesworthy research. Although this intervention was designed to strongly motivate participants to change their behavior, it had little impact on participants' decisions. The only impact of our experimental conditions is the

fact that pressure towards salesworthy research led people to increase their preference for reporting positive results. However, we did not find any impact of the intervention on the preference for randomized control experiments, or on the use of high sample sizes.

## General discussion

In two experiments, we show that people can recognize good signs of epistemic credibility: they prefer reporting experiments showing strong methodological features, such as a high sample size and the use of a control group. They are, however, also attracted towards positive results, and more likely to report on articles that strengthen their own pre-existing beliefs. These results mirror other studies showing human's vulnerability to positive result bias and confirmation bias [10, 11]. In order to counteract those biases, we used three different kinds of interventions in our two experiments, and found limited impact of our interventions on participants' behavior. The two framing nudges that we used to draw attention to the importance of null results and to promote scientifically accurate journalism did not impact participants' reporting practices. Even stronger incentives such as a pro-science work culture supported by colleagues and hierarchical superiors failed to increase participants' reporting behavior towards more accurate research. On the contrary, we found that work culture can have a negative impact, as participants expected that social pressure towards salesworthy research would lead them to show a stronger bias for reporting positive results.

While our results suggest that influencing participants' behavior towards high-quality scientific reporting is hard, several limitations should be noted. The first limitation of our study obviously lies in the fact that research integrity is linked with behavior, and our studies ask participants about hypothetical choices. While this use of hypothetical vignettes was mostly a matter of convenience, it reveals important factors that are likely at stake in real-world decisions. In our mind, asking about hypothetical scenarios sets an upper bound to the impact of nudges. Real behavior is likely to be even more multifaceted, and to have multiple causes. It will consequently be harder to influence real behavior than choices made in hypothetical situations. In this context, the fact that we found mostly null results is important, as it shows that such subtle interventions are unlikely to have much of an impact in the real world.

A second limitation stems from our choice to ask participants to imagine that they are science journalists, even though it is unlikely that they are or will become journalists in their real life. While we felt that such role-playing would be natural for most participants, some may consider this setting to be artificial. However, we think that this limitation is counterbalanced by more important methodological advantages. First, we wanted to estimate whether appealing to social roles (as in the *Social role* nudge of Study 1) may have a positive impact on science communication. We assumed that these social roles could be generalized to different social profiles where people have to communicate information (such as teachers, scientists, or science communicators). As such, asking people to imagine that they were journalists was essential to our design. Second, we chose to put participants in the shoes of a serious professional in order to avoid other factors that are strongly linked to more common contexts of sharing information with friends (e.g. via a social media application). Indeed, in informal contexts, one may be tempted to share more surprising, or funny, or personal-related information. While these factors are important and should be studied in their own right, they would have added additional noise and would have diminished our ability to detect any effect.

A third major limitation lies in our sampling procedure. Our participants were more educated than the general population in the UK and the US. According to a 2021 study by the American census bureau, only 48% of the American population aged over 25 have completed some college degree, while around 75% of our American sample have completed some college

degree [39]. According to the 2021 census, only 34% of the English and Welsh population aged over 16 have completed some college degree, compared to 63% of our sample [40]. It is likely that the preference for RCT and high sample sizes that we found would be lower in a more representative sample. Still, studies from Mturk find strong generalizability when replicated in probabilistic surveys [26].

Overall, our research suggests that participants may already be motivated to transmit reliable information, as shown in the importance of RCT and high sample sizes. Our results suggest that while participants understand that positive results are more newsworthy, they may be unable to see the risks of overreporting positive results. Our results highlight the fact that nudges of the sort we tested are unlikely to counteract epistemic vices and hence to positively contribute to the transmission of reliable scientific information. Exploration of more efficient strategies and active promotion of science education are still needed.

## Supporting information

**S1 File.**
(DOCX)

## Acknowledgments

This article benefitted from comments made by three anonymous reviewers. Reviewer 1 made suggestions that were outstanding in their quality and depth. We are grateful for their help, and for the time they have taken to write such a thoughtful review.

## Author Contributions

**Conceptualization:** Aurélien Allard, Christine Clavien.

**Data curation:** Aurélien Allard.

**Formal analysis:** Aurélien Allard.

**Funding acquisition:** Christine Clavien.

**Investigation:** Aurélien Allard.

**Methodology:** Aurélien Allard, Christine Clavien.

**Supervision:** Christine Clavien.

**Writing – original draft:** Aurélien Allard.

**Writing – review & editing:** Aurélien Allard, Christine Clavien.

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
