## [Decision Letter · Decision Letter 0]

13 Jun 2023

PONE-D-23-07977Nudging accurate scientific communicationPLOS ONE

Dear Dr. Clavien,

Thank you for submitting your manuscript to PLOS ONE. After careful consideration, we feel that it has merit but does not fully meet PLOS ONE’s publication criteria as it currently stands. Therefore, we invite you to submit a revised version of the manuscript that addresses the points raised during the review process.

Reviewers have found it difficult to understand several aspects of your manuscript, including the hypothesis and methods. I kindly ask you to carefully consider their comments to improve the clarity of the article and to address their concerns.

We look forward to receiving your revised manuscript.

Kind regards,

Alberto Molina Pérez, Ph.D.

Academic Editor

PLOS ONE

3. PLOS requires an ORCID iD for the corresponding author in Editorial Manager on papers submitted after December 6th, 2016. Please ensure that you have an ORCID iD and that it is validated in Editorial Manager. To do this, go to ‘Update my Information’ (in the upper left-hand corner of the main menu), and click on the Fetch/Validate link next to the ORCID field. This will take you to the ORCID site and allow you to create a new iD or authenticate a pre-existing iD in Editorial Manager. Please see the following video for instructions on linking an ORCID iD to your Editorial Manager account: https://www.youtube.com/watch?v=_xcclfuvtxQ.

Reviewers' comments:

Reviewer's Responses to Questions

**Comments to the Author**

1. Is the manuscript technically sound, and do the data support the conclusions?

Reviewer #1: Yes

Reviewer #2: Partly

Reviewer #3: Yes

2. Has the statistical analysis been performed appropriately and rigorously? 

Reviewer #1: Yes

Reviewer #2: Yes

Reviewer #3: Yes

3. Have the authors made all data underlying the findings in their manuscript fully available?

Reviewer #1: Yes

Reviewer #2: Yes

Reviewer #3: Yes

4. Is the manuscript presented in an intelligible fashion and written in standard English?

Reviewer #1: Yes

Reviewer #2: Yes

Reviewer #3: Yes

5. Review Comments to the Author

Reviewer #1: Thank you for the opportunity to review this paper. I read it with great interest. The topic is fascinating and, in light of the importance that social media nowadays plays in the sharing and spread of scientific (and other) news, the questions that the authors address in this paper are timely and important. The authors report findings from two large, pre-registered empirical studies, which is great. I also wholeheartedly agree with the authors that publication and discussion of null results emanating from scientific research is often as important as that of positive results. Before the paper gets published, however, I think the paper could be improved in a number of ways. I list my main, more general suggestions and remarks first, and smaller points after that.

The more general suggestions, remarks, and questions:

On the whole, the introduction to the paper is very well written and it motivates clearly the need for empirical studies that the authors conducted. However, as I read the paper, I often found myself flicking back and forth through the pages, and I had to consult the pre-registration document as well as the supplementary PDF files that the authors uploaded onto the Open Science Framework (OSF) database, in order to better understand the hypotheses that the authors set out to address and how their empirical studies did that. I think it’d be good to either have a single supplementary information document to accompany this paper and to make clear and direct references to that document from the main text to help a reader connect the dots, or to add references from the main text to the PDFs already uploaded onto the OSF, along with some additional guidance in the main text. This general blathering aside, below are more specific points in that regard.

1. In the introduction, when setting the scene for the subsequent discussion of empirical studies, the authors discuss the positive results bias and the confirmation bias in the dissemination of research results. This gives the impression that the nudges that the authors subsequently investigate in their empirical studies are meant to counteract these both forms of bias. However, if I understood the rest of the paper correctly, the authors assess the nudges only with respect to their effectiveness to counteract the positive results bias, but not the confirmation bias. Regarding the latter, the empirical studies were only meant to reveal (or confirm) the prevalence of the confirmation bias. It’d be good to make that clearer and more explicit in the introduction.

2. In the introduction the authors discuss the prospect of nudging people towards disseminating rigorous scientific results. However, in the context of the whole paper, the meaning of “rigorous” is somewhat obscure. The question of what counts as rigorous comes up, for example, when reading examples of vignettes that the authors used in their empirical studies. In all the provided examples (the one on p. 8 in the main text and the additional ones in the “Materials Main vignettes” PDF document on OSF), the sample size criterion favours one option while the control group criterion (that is, the use of randomized control trials) favours another. In such cases it is ambiguous which of the two presented options would reveal a choice to disseminate rigorous science. Presumably rigorous science is associated with larger sample sizes, the presence of control groups, no bias towards positive results, and no bias towards a specific research institution on the whole, that is, everything else being equal. a) It would be good to explain that somewhere early on in the text, perhaps where the authors discuss their experimental design and hypothesized predictions. I found the authors’ discussion of their predictions listed in the pre-registration document associated with the first study very useful for that matter. b) I think it’d be good to present those predictions somewhere early on in the main text too.

c) It might also be helpful to explain to a reader that, despite their being vignettes where what counts as a choice of reporting rigorous science is ambiguous, these cases are counterbalanced with cases where that question has a clear answer (large sample size + the presence of a control group vs. small sample size + no control group). As a result, we should expect rigorous science to be associated with a bias for larger sample sizes and for the presence of a control group across the board, that is, across all participants and all possible vignettes. It took me a while to appreciate that when thinking about the authors’ discussed results. I think that such slight (explicit) guidance for a reader’s thoughts would make it easier for one to appreciate the authors’ findings.

d) Lastly, to better understand the extent to which lay people’s choices diverge from the ideal, it would be nice to present a Figure that would show how participants’ actual choices compared to the choices of an ideal, unbiased reporter. Such a Figure would also visualize the actual sizes of some of the discussed effects that the authors subsequently report from their performed regressions. I don’t insist on the authors necessarily adding such a Figure to their paper, but I think that a Figure like this would be very useful for general clarity and for appreciation of the effect sizes that are at play.

3. In the discussion of how the authors determined their target sample sizes in the paragraph preceding the section “Materials” on p. 7 (lines 141–156), it’d be good if the authors could add a sentence or two on what simulations they performed, i.e., what statistical methods they used (instead of, or in addition to, pointing a reader to the R code uploaded onto OSF).

4. In the section “Materials,” where the authors present an example of a vignette they used, it’d be very good point a reader towards a supplementary information document (or the “Materials Main vignettes” PDF on OSF) to look at vignettes for each of the 5 scenarios that participants were presented (medical drug, psychological therapy, etc.). I also strongly recommend to add a bit of variety to the examples presented in the “Materials Main vignettes” PDF. Presently all the presented examples in this supplementary document pit a study with a small sample + no control group + positive result vs. a study with a large sample + control group + null result. It would be great to replace some of these with the following two cases:

a) small sample + control group + positive result vs. large sample + no control group + null result;

b) small sample + no control group + null result vs. large sample + control group + positive result.

One can figure out how the various vignettes looked like from going through the “Materials_First_Experiment_Science_Journalism” PDF on OSF. However, one needs to do quite some work inspecting the code that the authors used to put the various bits of text together to form the presented sentences in their vignettes.

5. There’s a sentence in the presented vignette on p. 8 that says “After 5 days, around 50% of the participants had seen their condition improve in both the placebo group and the treatment group, since their blood pressure decreased.” When I read the examples of other vignettes in the “Materials Main vignettes” PDF on OSF, I noticed that in the example associated with the new medical drug condition there, the terms used were not “the placebo group” and “the treatment group,” but “the placebo group” and “the control group.” The latter pair is confusing because usually we refer to a placebo group and a treatment group, or a control group and a treatment group, but not to a placebo group and a control group. When inspecting the code that the authors used to generate these vignettes (presented in the “Materials_First_Experiment_Science_Journalism” PDF on OSF) I noticed that there was indeed this “error” in the code. It would be good if the authors would make a point about that and perhaps re-run their regressions excluding the participants who were exposed to the vignette that used the expressions “the placebo group” and “the control group” to contrast the two groups in the same sentence. If the results remain largely unchanged, it might be worth simply mentioning that in a footnote and leave the rest of the paper as is.

6. Strictly speaking, the preference ranking that the authors elicited from participants’ choices of which study to report as discussed on p. 9 (lines 191–200) gives an ordinal measure of preference. For such categorical, ordinal data of a dependent variable, it is often advisable to perform ordinal logistic regressions instead of straightforward linear regressions. However, as far as is known to me, the scientific community is in two ways regarding this. Some people insist on ordinal logistic regressions, others are happy with using simple linear regressions. I’m not going to pick a side here and do not insist that the authors perform ordinal logistic regressions. (In fact, interpreting results from ordinal logistic regressions is a nightmare when it comes to interaction terms and for that reason some people advise against their use.) That said, if the authors could add a sentence or two to justify their decision to perform simple linear regressions (or perhaps add a Figure plotting the data that would support their choice), that would be a good thing to do. One possibility is to reference some previous studies that performed simple linear regressions on similar data.

7. Regarding the exclusion of participants for data analysis on pp. 11-12. The authors mention that they excluded participants who “copy-pasted citations from the internet” when providing reasons for their choices (lines 261–262). It’d be good if the authors could explain how they identified statements as copy-pasted.

8. It’d be good if somewhere in the text the authors reported summary demographic statistics of their recruited participants.

9. I think the last few sentences in the conclusion are a tad too strong. These results show that the specific nudges that the authors tested were not very effective. This doesn’t mean that nudges in general won’t work. Perhaps it is possible to design other types of nudge that would work?

Smaller points:

10. Introduction, second paragraph, last sentence: “accurate transmission of information” (lines 40–41). I think it’s more fitting here to say “transmission of accurate information.” That would also make it consistent with the expression used in the sentence preceding this one.

11. Introduction, the last sentence of the first paragraph on p. 4: “We chose to study the impact of nudges, or minimal interventions that try to influence participants in a desirable direction without changing the incentive structure faced by the participants (Thaler & Sunstein, 2021)” (lines 70-72). That is all fair, but it would be good if the authors could add a sentence or two on why it is particularly important or useful to consider nudge interventions as opposed to other types of intervention, e.g., those that would indeed change the “incentive structure” when it comes to dissemination of scientific news. On the one hand, if everyone agrees on what constitutes good and bad research, why not simply change the “incentive structure” itself? On the other hand, perhaps changing the incentive structure is not always possible or is too costly?

12. In the introduction, the authors present a formal name for the first type of nudge that they set out to investigate: the Attention to the Null Hypothesis nudge. It’d be good to present a formal name for the second type of nudge as well. For example, the Social Norm nudge. That would make it easier to follow the discussion in what follows.

13. Introduction, last paragraph, first sentence: “we study whether it is possible to steer people towards more rigorous research” (line 120). I think it’s more fitting here to say “towards reporting more rigorous research” or “towards propagating more rigorous research.”

14. Methods, last paragraph on p. 9: “participants had to report on the hypothesis that some intervention had a positive impact” (lines 211–212). It’d be good to make this clearer: participants had to report their initial intuition concerning the hypothesis that they were tasked to assess. Similarly in the sentence that follows this one. Also on the following p. 10: “participants then had to give plausibility ratings” (line 224). It is probably more accurate to say “their initial intuition concerning the plausibility of their assessed hypotheses” (or similar).

15. Materials, last paragraph on p. 9: “a drug was improving some illness” (line 212). Bad wording: the drug was not improving an illness, but was effective in treating the illness (or something similar).

16. Concerning the elicitation of participants’ initial intuitions regarding the assessed hypotheses on p. 10 (lines 224–229), it’d be good to give a bit more detail. a) Provide all five options that participants had to choose from in the Competing hypotheses treatment (otherwise it’s unclear how statements other than at the two extremes may have looked like). b) Explain what the options were in the Positive hypothesis only treatment. Alternatively, the authors could point a reader towards this information in supplementary documents (e.g., to the relevant page in one of the PDF documents on OSF).

17. The last paragraph preceding the section “Results” on p. 11. Again, it’d be good point a reader towards supplementary documents to find the exact wordings that were used to elicit these additional data in surveys.

18. Results, Table 2 on p. 13, 4th line row: “Positive Hypothesis.” Should this say “Positive Hypothesis Only”?

19. Results, first paragraph on p. 13. Going back to some of the points I made earlier, I think this is another place where it’d be useful to remind a reader what the various predictions for the interaction of the Positive Hypothesis Only variable with other variables were.

20. Second paragraph on p. 25: “This intervention was quite strong” (lines 504–505). This wording isn’t quite clear. Perhaps there’s a way to rephrase this.

21. Conclusion, one but last paragraph on p. 26: “Participants from Prolific and Amazon Mechanical Turk tend to be more educated than the general population” (lines 536–537). It’d be nice to add a reference here if possible.

Really minor points and typos:

22. Materials, first paragraph, first sentence: “Inspired by Bottesini et al., 2021” (line 159). “I” should not be capitalized.

23. Materials, first paragraph on p. 8: “Microfinance on poverty” (line 163). “M” should probably not be capitalized.

24. Materials, p. 8: “For instance, in the drug condition” (line 174). For clarity, perhaps better to use the full name given to this condition earlier on: “the new medical drug condition.”

25. Materials, p. 10: “or two both hypotheses” (line 225). Probably should say “to” instead of “two.”

26. Results, p. 12, last paragraph: “in conformity with our predictions” (line 279). Perhaps worth adding “in conformity with our predictions and results from previous studies” (to reflect the earlier discussion in the introduction).

27. Figure 1 and the Tables were not explicitly referenced in the text. I think it’d be good to add explicit references to them. Also, it’d be good to expand the caption of Figure 1 to explain it in more detail, for example, what is on the y-axis.

28. First paragraph on p. 24: “only one of the effect” (line 487). This should be plural: “effects.”

Reviewer #2: The paper has it’s strength in the data. However, the empirical design is hard to follow. There are two experiments that have been carried out at different point of times. One factor is the same, further variations take place (Yale vs Princeton), which are not explained. Each participant read one text so there is need to vary affiliations.

Also, asking participants to imagine they were journalists Arena Problematik and Shirley be Diskusses in the limitations more thoroughly. Nudging accurate science and be a journalist is not always in the Dame line.

One limitation is that none descriptives are available about the samples.

Reviewer #3: This is a technically very good article, well written and with a solid experimental design. From this point of view, there would be no reason to criticize it. But there are reasons, in my opinion, to question a central aspect of the experimental design.

The authors start from a clear and correct statement:

"Ordinary citizens are important conveyors of scientific information, in their decision to share information with friends, family, and, in the case of social media, even with random stranger"

Immediately afterwards, they state that

"In this paper, we study several factors influencing non-specialists’ treatment of scientific information"

In particular, it focuses on two very important biases: a positive outcome bias, and a confirmation bias.

In the end they conclude that people know how to recognize the signs of epistemic credibility, but they are "also attracted towards positive results, and more likely to report on articles that strengthen their own pre-existing beliefs. [...]. nudges are not enough to counteract epistemic vices".

Among the limitations, the authors highlight the fact that the experiment is hypothetical. And this is where the problem lies. To find out whether laypeople, as transmitters of scientific information, suffer from the two biases mentioned above, was it necessary to assume that the participants had to imagine that they were science journalists who had to select scientific studies to report in their next article? It seems to me a very forced artifice not explained in the article. It is not just that people lack experience in these tasks, but that the authors could have imagined an experimental design in which people, for example, obtain scientific information -by reading it, watching documentaries, etc.- and transmit it to others, to see to what extent they transmit the information in a way that reinforces their beliefs and positive results. The idea that laypeople are science journalists seems to me to be inadequate to "study several factors influencing non-specialists’ treatment of scientific information".

Moreover, it should have been said that "nudges are not enough to counteract epistemic vices" in an overly contrived context in which people act as if they were science journalists. Would they have worked in a more realistic context in which laypeople convey information without imagining that they are science journalists? Would they have worked among real science journalists? We don't know.

Now, if one accepts that this design is good enough, the article can be published as it is.

6. PLOS authors have the option to publish the peer review history of their article (what does this mean?). If published, this will include your full peer review and any attached files.

Reviewer #1: No

Reviewer #2: No

Reviewer #3: No

---

## [Author Response · Author response to Decision Letter 0]

26 Jul 2023

We have uploaded a file with our point by point responses to reviewers. Here is a copy-past of that file.

We wish to thank all reviewers for their outstanding comments which helped us clarify and improve the manuscript. This version of the manuscript is now much more developed.

Reviewer #1

The more general suggestions, remarks, and questions:

On the whole, the introduction to the paper is very well written and it motivates clearly the need for empirical studies that the authors conducted. However, as I read the paper, I often found myself flicking back and forth through the pages, and I had to consult the pre-registration document as well as the supplementary PDF files that the authors uploaded onto the Open Science Framework (OSF) database, in order to better understand the hypotheses that the authors set out to address and how their empirical studies did that. I think it’d be good to either have a single supplementary information document to accompany this paper and to make clear and direct references to that document from the main text to help a reader connect the dots, or to add references from the main text to the PDFs already uploaded onto the OSF, along with some additional guidance in the main text.

In the introduction, when setting the scene for the subsequent discussion of empirical studies, the authors discuss the positive results bias and the confirmation bias in the dissemination of research results. This gives the impression that the nudges that the authors subsequently investigate in their empirical studies are meant to counteract these both forms of bias. However, if I understood the rest of the paper correctly, the authors assess the nudges only with respect to their effectiveness to counteract the positive results bias, but not the confirmation bias. Regarding the latter, the empirical studies were only meant to reveal (or confirm) the prevalence of the confirmation bias. It’d be good to make that clearer and more explicit in the introduction.

This interpretation is correct. We had to make choices among hypotheses to test, and have made it more explicit in the introduction by adding the following sentence on L. 70: "Our practical goal was to mitigate the influence of the positive results bias, and to enhance preferences for more rigorous research."

Moreover, as suggested, we have also added a single supplementary file that includes one appendix describing the vignettes that we used.

In the introduction the authors discuss the prospect of nudging people towards disseminating rigorous scientific results. However, in the context of the whole paper, the meaning of “rigorous” is somewhat obscure. The question of what counts as rigorous comes up, for example, when reading examples of vignettes that the authors used in their empirical studies. In all the provided examples (the one on p. 8 in the main text and the additional ones in the “Materials Main vignettes” PDF document on OSF), the sample size criterion favours one option while the control group criterion (that is, the use of randomized control trials) favours another. In such cases it is ambiguous which of the two presented options would reveal a choice to disseminate rigorous science. Presumably rigorous science is associated with larger sample sizes, the presence of control groups, no bias towards positive results, and no bias towards a specific research institution on the whole, that is, everything else being equal. a) It would be good to explain that somewhere early on in the text, perhaps where the authors discuss their experimental design and hypothesized predictions. I found the authors’ discussion of their predictions listed in the pre-registration document associated with the first study very useful for that matter. b) I think it’d be good to present those predictions somewhere early on in the main text too.

We have clarified in the introduction that our objective (and related prediction) was to use soft interventions (nudges) in order to lead people to prefer experiments with a high sample size, with a control group, and to be less biased in favour of positive results. We also make it explicit that this is what we mean by “rigorous” in this article.

c) It might also be helpful to explain to a reader that, despite their being vignettes where what counts as a choice of reporting rigorous science is ambiguous, these cases are counterbalanced with cases where that question has a clear answer (large sample size + the presence of a control group vs. small sample size + no control group). As a result, we should expect rigorous science to be associated with a bias for larger sample sizes and for the presence of a control group across the board, that is, across all participants and all possible vignettes. It took me a while to appreciate that when thinking about the authors’ discussed results. I think that such slight (explicit) guidance for a reader’s thoughts would make it easier for one to appreciate the authors’ findings.

Thanks for the comment. We have tried to clarify this important aspect, by adding in the main text an explanatory paragraph and a citation of a vignette where participants could find both the high sample size condition and the RCT in the same experiments. We have tried to make it clearer that each factor was orthogonally manipulated.

d) Lastly, to better understand the extent to which lay people’s choices diverge from the ideal, it would be nice to present a Figure that would show how participants’ actual choices compared to the choices of an ideal, unbiased reporter. Such a Figure would also visualize the actual sizes of some of the discussed effects that the authors subsequently report from their performed regressions. I don’t insist on the authors necessarily adding such a Figure to their paper, but I think that a Figure like this would be very useful for general clarity and for appreciation of the effect sizes that are at play.

We have considered the option of presenting such a figure, however, decided not to do so because it could be misleading. Indeed, we find it somewhat arbitrary to decide exactly how much people should be swayed in one direction or the other. We think that any answer between "should mostly mention study A" to "should only mention study A" could count as rational but a figure with linear outputs may not give that impression. In other words, we are mostly assuming that the impact of RCT and sample size should sum to at least 1, and that the impact of positive results should be 0. This is difficult to illustrate.

In the discussion of how the authors determined their target sample sizes in the paragraph preceding the section “Materials” on p. 7 (lines 141–156), it’d be good if the authors could add a sentence or two on what simulations they performed, i.e., what statistical methods they used (instead of, or in addition to, pointing a reader to the R code uploaded onto OSF).

We added a brief description of the simulations. However, we found it hard to describe the simulations while keeping it short enough to avoid distracting readers. We hope that this modified version will be helpful to our readers.

In the section “Materials,” where the authors present an example of a vignette they used, it’d be very good point a reader towards a supplementary information document (or the “Materials Main vignettes” PDF on OSF) to look at vignettes for each of the 5 scenarios that participants were presented (medical drug, psychological therapy, etc.). I also strongly recommend to add a bit of variety to the examples presented in the “Materials Main vignettes” PDF. Presently all the presented examples in this supplementary document pit a study with a small sample + no control group + positive result vs. a study with a large sample + control group + null result. It would be great to replace some of these with the following two cases:

a) small sample + control group + positive result vs. large sample + no control group + null result;

b) small sample + no control group + null result vs. large sample + control group + positive result.

One can figure out how the various vignettes looked like from going through the “Materials_First_Experiment_Science_Journalism” PDF on OSF. However, one needs to do quite some work inspecting the code that the authors used to put the various bits of text together to form the presented sentences in their vignettes.

That's a great suggestion. We created more varied vignettes in the Appendix A in the supplementary file, and we added the reference to the vignettes in the main text.

There’s a sentence in the presented vignette on p. 8 that says “After 5 days, around 50% of the participants had seen their condition improve in both the placebo group and the treatment group, since their blood pressure decreased.” When I read the examples of other vignettes in the “Materials Main vignettes” PDF on OSF, I noticed that in the example associated with the new medical drug condition there, the terms used were not “the placebo group” and “the treatment group,” but “the placebo group” and “the control group.” The latter pair is confusing because usually we refer to a placebo group and a treatment group, or a control group and a treatment group, but not to a placebo group and a control group. When inspecting the code that the authors used to generate these vignettes (presented in the “Materials_First_Experiment_Science_Journalism” PDF on OSF) I noticed that there was indeed this “error” in the code. It would be good if the authors would make a point about that and perhaps re-run their regressions excluding the participants who were exposed to the vignette that used the expressions “the placebo group” and “the control group” to contrast the two groups in the same sentence. If the results remain largely unchanged, it might be worth simply mentioning that in a footnote and leave the rest of the paper as is.

Thanks for catching this! We are very sorry about the error. Fortunately, the only time we refer to the placebo group AND the control is in the sentence: "around 50% of the participants had seen their condition improve in both the control group and the placebo group, since their blood pressure decreased. The researchers concluded that the drug was not working". We do feel that participants would have correctly interpreted this reference to the "control" group as a typo, and that they would correctly have concluded that there was no difference between the treatment group and the placebo group. We thus believe that the main results reported in the text are not affected.

However, we agree with the reviewer that the reader should be informed of this, and added a section on robustness checks in the Results section of Study 1. We have re-run our analyses excluding participants in the "Drug" condition. Our results are unaffected. The experimental features remain significant (with qualitatively similar coefficients), the nudges remain non-significant, the confirmation bias remains significant (all in both Study 1 and Study 2), and the interaction between Pressure towards interesting research and preference for positive results remains significant in Study 2.

We report the additional analyses in appendix B.

Strictly speaking, the preference ranking that the authors elicited from participants’ choices of which study to report as discussed on p. 9 (lines 191–200) gives an ordinal measure of preference. For such categorical, ordinal data of a dependent variable, it is often advisable to perform ordinal logistic regressions instead of straightforward linear regressions. However, as far as is known to me, the scientific community is in two ways regarding this. Some people insist on ordinal logistic regressions, others are happy with using simple linear regressions. I’m not going to pick a side here and do not insist that the authors perform ordinal logistic regressions. (In fact, interpreting results from ordinal logistic regressions is a nightmare when it comes to interaction terms and for that reason some people advise against their use.) That said, if the authors could add a sentence or two to justify their decision to perform simple linear regressions (or perhaps add a Figure plotting the data that would support their choice), that would be a good thing to do. One possibility is to reference some previous studies that performed simple linear regressions on similar data.

We have now justified our choice in a new “Statistical Analysis” section, by appealing to 1) simplicity, 2) past use within the scientific literature, and 3) the fact that psychological variables often seem to have linear impact on other variables, leaving open the possibility that Likert-like scale are indeed interpreted by participants as interval scales (with the same distance between any two contiguous response options).

Regarding the exclusion of participants for data analysis on pp. 11-12. The authors mention that they excluded participants who “copy-pasted citations from the internet” when providing reasons for their choices (lines 261–262). It’d be good if the authors could explain how they identified statements as copy-pasted.

In fact we trusted our intuitions to identify what were copy-pasted citations from the internet. We classified as copy-pasted any fully-formed sentence that did not make any sense in the context of the experiment. We now make it more explicit in the text and provide one example of a copy-pasted statement ("Elements of Bader's theory of atoms in molecules are combined with density-functional theory to provide an electron-preceding perspective on the deformation of materials.").

It’d be good if somewhere in the text the authors reported summary demographic statistics of their recruited participants.

We now provide full summary demographic statistics in the main text.

I think the last few sentences in the conclusion are a tad too strong. These results show that the specific nudges that the authors tested were not very effective. This doesn’t mean that nudges in general won’t work. Perhaps it is possible to design other types of nudge that would work?

The wording was indeed a tad too strong. We re-wrote these sentences. 

Smaller points:

Introduction, second paragraph, last sentence: “accurate transmission of information” (lines 40–41). I think it’s more fitting here to say “transmission of accurate information.” That would also make it consistent with the expression used in the sentence preceding this one.

Thanks! We agree, and made the change accordingly. 

Introduction, the last sentence of the first paragraph on p. 4: “We chose to study the impact of nudges, or minimal interventions that try to influence participants in a desirable direction without changing the incentive structure faced by the participants (Thaler & Sunstein, 2021)” (lines 70-72). That is all fair, but it would be good if the authors could add a sentence or two on why it is particularly important or useful to consider nudge interventions as opposed to other types of intervention, e.g., those that would indeed change the “incentive structure” when it comes to dissemination of scientific news. On the one hand, if everyone agrees on what constitutes good and bad research, why not simply change the “incentive structure” itself? On the other hand, perhaps changing the incentive structure is not always possible or is too costly?

We have clarified our perspective on nudges in this paragraph. In our view, nudges should be considered as a good first option, since they are easy to implement and are not costly. However, nudges should not be considered as opposed to other stricter interventions. Often, they can be complementary. 

In the introduction, the authors present a formal name for the first type of nudge that they set out to investigate: the Attention to the Null Hypothesis nudge. It’d be good to present a formal name for the second type of nudge as well. For example, the Social Norm nudge. That would make it easier to follow the discussion in what follows.

Thank you for the suggestion. We agree and have re-named the other nudge the Social Role nudge.

Introduction, last paragraph, first sentence: “we study whether it is possible to steer people towards more rigorous research” (line 120). I think it’s more fitting here to say “towards reporting more rigorous research” or “towards propagating more rigorous research.”

Thanks! Corrected.

Methods, last paragraph on p. 9: “participants had to report on the hypothesis that some intervention had a positive impact” (lines 211–212). It’d be good to make this clearer: participants had to report their initial intuition concerning the hypothesis that they were tasked to assess. Similarly in the sentence that follows this one. Also on the following p. 10: “participants then had to give plausibility ratings” (line 224). It is probably more accurate to say “their initial intuition concerning the plausibility of their assessed hypotheses” (or similar).

Thanks! Corrected.

Materials, last paragraph on p. 9: “a drug was improving some illness” (line 212). Bad wording: the drug was not improving an illness, but was effective in treating the illness (or something similar).

Thanks! Corrected.

Concerning the elicitation of participants’ initial intuitions regarding the assessed hypotheses on p. 10 (lines 224–229), it’d be good to give a bit more detail. a) Provide all five options that participants had to choose from in the Competing hypotheses treatment (otherwise it’s unclear how statements other than at the two extremes may have looked like). b) Explain what the options were in the Positive hypothesis only treatment. Alternatively, the authors could point a reader towards this information in supplementary documents (e.g., to the relevant page in one of the PDF documents on OSF).

Thanks, we have provided additional details here.

The last paragraph preceding the section “Results” on p. 11. Again, it’d be good point a reader towards supplementary documents to find the exact wordings that were used to elicit these additional data in surveys.

We have redirected the reader towards these.

Results, Table 2 on p. 13, 4th line row: “Positive Hypothesis.” Should this say “Positive Hypothesis Only”?

Ok. Corrected.

Results, first paragraph on p. 13. Going back to some of the points I made earlier, I think this is another place where it’d be useful to remind a reader what the various predictions for the interaction of the Positive Hypothesis Only variable with other variables were.

Thank you for your suggestion. We agree that this would improve the clarity of the results section, and have added this reminder accordingly.

Second paragraph on p. 25: “This intervention was quite strong” (lines 504–505). This wording isn’t quite clear. Perhaps there’s a way to rephrase this.

We have reformulated this sentence.

Conclusion, one but last paragraph on p. 26: “Participants from Prolific and Amazon Mechanical Turk tend to be more educated than the general population” (lines 536–537). It’d be nice to add a reference here if possible.

Thank you for this suggestion. We based this information on our experience using these platforms, but we couldn’t find a recent reference for Prolific Academic. So we actually compared the educational attainment from our sample with representative figures from the US and the UK. Note that the comparison is not perfectly adequate; we found accessible data only for England and Wales, and not for the UK as a whole (England and Wales still comprise about 90% of the UK population, however). Moreover, the categories used by the censuses are different from the ones used in our survey. Despite these measurement errors, our figures show that participants from both Mturk and Prolific are much more educated than the general population, and we think that this constitutes useful information for the reader to keep in mind.

Really minor points and typos:

Materials, first paragraph, first sentence: “Inspired by Bottesini et al., 2021” (line 159). “I” should not be capitalized.

Thanks! Corrected.

Materials, first paragraph on p. 8: “Microfinance on poverty” (line 163). “M” should probably not be capitalized.

Thanks! Corrected.

Materials, p. 8: “For instance, in the drug condition” (line 174). For clarity, perhaps better to use the full name given to this condition earlier on: “the new medical drug condition.”

Thanks. Changes made.

Materials, p. 10: “or two both hypotheses” (line 225). Probably should say “to” instead of “two.”

Thanks! Corrected.

Results, p. 12, last paragraph: “in conformity with our predictions” (line 279). Perhaps worth adding “in conformity with our predictions and results from previous studies” (to reflect the earlier discussion in the introduction).

Agreed. Change made.

Figure 1 and the Tables were not explicitly referenced in the text. I think it’d be good to add explicit references to them. Also, it’d be good to expand the caption of Figure 1 to explain it in more detail, for example, what is on the y-axis.

Thanks for this suggestion! We have referenced the tables and figure, and clarified the meaning of figure 1.

First paragraph on p. 24: “only one of the effect” (line 487). This should be plural: “effects.”

Corrected as well.

Many thanks again for all your comments and your detailed reading of our article.

Reviewer #2

The paper has it’s strength in the data. However, the empirical design is hard to follow. There are two experiments that have been carried out at different point of times. One factor is the same, further variations take place (Yale vs Princeton), which are not explained. Each participant read one text so there is need to vary affiliations.

We agree that the design of the experiment and the way vignettes were presented to our participants was not explained in a clear way in the first version of our paper. We now have made several improvements, notably be providing two examples in the text and by adding an explanatory Appendix (A). We hope that it will help the readers. 

Also, asking participants to imagine they were journalists Arena Problematik and Shirley be Diskusses in the limitations more thoroughly. Nudging accurate science and be a journalist is not always in the Dame line.

Regarding our choice of making participants imagine that they were science journalists, we agree that it may be considered as a limitation and have added a § in the conclusion to explain our choice. We agree that our participants are unlikely to be or become science journalists in their real life. However this limitation is counterbalanced by more important methodological advantages. It helps avoiding important confounding factors. To make it more explicit, we have added the following paragraph in our conclusion:

“A second limitation stems from our choice to ask participants to imagine that they are science journalists, even though it is unlikely that they are or will become journalists in their real life. While we felt that such role-playing would be natural for most participants, some may consider this setting to be artificial. We think however that this limitation is counterbalanced by more important methodological advantages. First, we wanted to estimate whether appealing to social roles may have a positive impact on science communication. We assumed that these social roles could be generalized to different social profiles where people have to communicate information (such as teachers, scientists, or science communicators). As such, asking people to imagine that they were journalists was essential to our design. Second, we chose to put participants in the shoes of a serious professional in order to avoid important confounding factors that are strongly linked to more common contexts of sharing information with friends (e.g. via a social media applications). Indeed, in informal contexts, one may be tempted to share more surprising, or funny, or personal-related information. While these factors are important, and should be studied in their own right, they would have added additional noise and would have diminished our ability to detect any effect.”

One limitation is that none descriptives are available about the samples.

Thanks for the comment! This was missing indeed. We have now added two tables describing the characteristics of our participants.

Reviewer #3: 

Among the limitations, the authors highlight the fact that the experiment is hypothetical. And this is where the problem lies. To find out whether laypeople, as transmitters of scientific information, suffer from the two biases mentioned above, was it necessary to assume that the participants had to imagine that they were science journalists who had to select scientific studies to report in their next article? It seems to me a very forced artifice not explained in the article. It is not just that people lack experience in these tasks, but that the authors could have imagined an experimental design in which people, for example, obtain scientific information -by reading it, watching documentaries, etc.- and transmit it to others, to see to what extent they transmit the information in a way that reinforces their beliefs and positive results. The idea that laypeople are science journalists seems to me to be inadequate to "study several factors influencing non-specialists’ treatment of scientific information".

This is a similar comment to reviewer 2, showing that we really need to address this issue explicitly in the paper. We agree that this is a limitation and have added a § in the concluding section to acknowledge it. However, we still think that such an artificial design is the price to pay in order to avoid more serious methodological difficulties, and to facilitate the implementation of our nudges. We have added the following explanatory paragraph in our conclusion:

“A second limitation stems from our choice to ask participants to imagine that they are science journalists, even though it is unlikely that they are or will become journalists in their real life. While we felt that such role-playing would be natural for most participants, some may consider this setting to be artificial. We think however that this limitation is counterbalanced by more important methodological advantages. First, we wanted to estimate whether appealing to social roles may have a positive impact on science communication. We assumed that these social roles could be generalized to different social profiles where people have to communicate information (such as teachers, scientists, or science communicators). As such, asking people to imagine that they were journalists was essential to our design. Second, we chose to put participants in the shoes of a serious professional in order to avoid important confounding factors that are strongly linked to more common contexts of sharing information with friends (e.g. via a social media applications). Indeed, in informal contexts, one may be tempted to share more surprising, or funny, or personal-related information. While these factors are important, and should be studied in their own right, they would have added additional noise and would have diminished our ability to detect any effect.”

Moreover, it should have been said that "nudges are not enough to counteract epistemic vices" in an overly contrived context in which people act as if they were science journalists. Would they have worked in a more realistic context in which laypeople convey information without imagining that they are science journalists? Would they have worked among real science journalists? We don't know. Now, if one accepts that this design is good enough, the article can be published as it is.

Thanks for the comment. It is true that our scenario is not fully realistic and have nuanced this sentence in the conclusion. Simplified models are however necessary to identify the specific effect of individual factors, and usually, the effects found in laboratory settings are stronger than in the real world. This is why we think that an absence of effect in our setting is a reliable sign that a positive effect of our interventions would be unlikely to happen in reality.

---

## [Editor Report · Decision Letter 1]

7 Aug 2023

Nudging accurate scientific communication

PONE-D-23-07977R1

Dear Dr. Clavien,

We’re pleased to inform you that your manuscript has been judged scientifically suitable for publication and will be formally accepted for publication once it meets all outstanding technical requirements.

Kind regards,

Alberto Molina Pérez, Ph.D.

Academic Editor

PLOS ONE

Additional Editor Comments (optional):

Just a very minor comment: page 5, lines 94-95, it may be preferable to say "large versus small" (rather than "small versus large").
---

## [Editor Report · Acceptance letter]

23 Aug 2023

PONE-D-23-07977R1 

Nudging accurate scientific communication 

Dear Dr. Clavien:

I'm pleased to inform you that your manuscript has been deemed suitable for publication in PLOS ONE. Congratulations! Your manuscript is now with our production department. 

Kind regards, 

on behalf of

Dr. Alberto Molina Pérez 

Academic Editor

PLOS ONE